# OpenReview forum: "Elephant in the Room: Unveiling the Pitfalls of Human Proxies in Alignment"
_ICLR.cc/2025/Conference — Submitted to ICLR 2025_

### Official Review · Reviewer_ZGTm · 2024-10-20

**Soundness:** 3
**Presentation:** 2
**Contribution:** 2
**Rating:** 3
**Confidence:** 4

**Summary:**

The paper raises an important question on whether current day alignment datasets are truly representative of human preferences. To address this question, the authors study to abstractions of human preferences - "Level-1 Proxy" (aka training data) and "Level-2 Proxy" (reward models).

Starting with Level-1 Proxy: The authors use HH-RLHF, a popular dataset, and offer a taxonomy to categorize the training data (Section 3.1), such as toxic responses, "inverted" responses (ie, the rejected response is actually better than the chosen respose), etc.
The authors then carefully re-label HH-RLHF using 4 human annotators, and find that a large chunk of the data falls under the above pitfalls.
Re-training on the cleaned data (CHH-RLHF) using DPO, they demonstrate immediate improvements in the model's "reward score" (Section 3.2.2, Table 2).

For Level-2 Proxy: The authors then study the impact that a reward model can have on the alignment process. Similar to Level-1, the authors start with a taxonomy of pitfalls (Section 4.1.2). However, if I understand correctly, the taxonomy does not seem to be used anywhere? (Ex:, “Score for Empty Responses” indicates when a reward model gives high scores to an empty response – how often does something like this happen?)

The authors then claim that current reward models are not adequate enough to be used for training aligned models (Section 4.2).
The authors define an accuracy metric to define the quality of a reward model, and assess a wide range of reward models (Table 4), demonstrating a wide range of accuracy scores.
The authors study the impact that a reward model can have on alignment algorithms - PPO and PRO – and find that suboptimal reward models lead to suboptimally aligned models.

**Strengths:**

* The re-labeled dataset CHH-RLHF is definitely a contribution for the community.
* The authors do an exhaustive list of experiments, although it is not clear how the set of models chosen for each experiment was decided on.

**Weaknesses:**

* My biggest question is what new information have we learned from this? I believe that the takeaways are that better data leads to better models, and better reward models lead to better aligned models. While a cleaned dataset is a contribution worth applauding, I don’t think there is anything else that the paper offers that we didn’t already know.

* The authors have a wide range of empirical results - but it is not clear how the choice of models used for each experiment was made.

* Figure 4: The authors claim that Starling-34B’s evaluations show good correlation with human evaluations (But is missing a correlation score?), and also shows that Starling-34B has the highest reward model accuracy (Table 4) and using Startling-34B as a reward model leads to a better model (Figure 6). Doesn’t this go entirely against the authors claim that current day reward models (level-2 proxy) are inadequate?

* Table 4: I don’t think assessing DPO for reward modeling is reasonable. There seems to be an assumption being made that the policy model from DPO would also serve as a good reward model – and the authors justify this assumption with the paper title of DPO (“Your language model is secretly a reward model”) – I don’t think this is reasonable, especially because prior work shows that policy models learn very different things from reward models (https://arxiv.org/abs/2405.19534).
The reason I mention this is that if you take out the DPO results in Table 4, the claims that the authors make (ie, reward models often do worse than 50/50 chance) is significantly weakened. My takeaways from Table 4 is that larger models lead to better reward modeling accuracy, and larger models (Starling-7B, 34B) do not seem to have the issues that the authors discuss.

**Questions:**

* Some details on HH-RLHF would be helpful. Datasize? How was it originally annotated?
Ie, in Figure 3, when 0.07% is said to be “Empty” – how many samples is that?

* Figure 4 – why did you include GPT-J? Also, how big of a model is GPT-J?
* Figure 4 - why is there no correlation score?
* Does the newly labeled data include annotations from all 4 annotators? If you have already gone through the effort to re-label the data, having this more granular information could allow for better alignment methods as opposed to binary (preferred vs. rejected) labels.
* What is the purpose of Figure 5?

---

> ### Author Response · Authors · 2024-11-25
> **Response to Reviewer ZGTm (Part 1)**
>
> We would like to thank the reviewer for the insightful and constructive comments. The following are our detailed responses regarding all major concerns. We hope the following responses can clarify the missing points and address these concerns.
>
> ## Addressing Weaknesses:
>
> **W1**. No new information. (I believe that the takeaways are that better data leads to better models, and better reward models lead to better aligned models. While a cleaned dataset is a contribution worth applauding, I don’t think there is anything else that the paper offers that we didn’t already know.)
>
> **A1**: This is the elephant we are discussing. Everyone seems to know better human proxies lead to better alignment with true human intentions; however, existing literature in alignment research is widely using bad human proxies without any verification. What's worse, the evaluation of various alignment algorithms is conducted using these bad human proxies, which makes it difficult to distinguish whether an alignment algorithm is indeed good or not. For example, RRHF[1] uses the same bad reward model, GPTJ, both for alignment optimization and evaluation. This makes their results untrustworthy. More trustworthy evaluation should be done with good human proxies, and the correlation between human evaluation and proxy evaluation should be provided for reference, like what we have presented in our paper.
> Besides, it is not always true that "better data leads to better models and better reward models lead to better aligned models". OpenAI [2] has discovered the upper bound by introducing the concept of Goodhart's law, which mainly states that if reward models perform beyond a certain upper bound, the alignment performance surprisingly decreases. We also emphasize this in Section 5. Different from OpenAI's work, what we explore in this work is actually the opposite side by pointing out that under-optimized reward models lead to misalignment. We want to raise researchers' attention to verify the quality of two levels of human proxies before usage, either for alignment optimization or evaluation. People have taken it for granted that better reward models lead to better performance and directly assume the reward models they use are indeed good, which is nevertheless not true. We unveil this neglected elephant in our work. Current researchers of alignment have presumed many things that do not exist. They presume the preference data they use indeed reflects humans' intentions. They presume the reward models they use are good enough for both alignment and evaluation without any verification. If someone checks all the alignment works in recent years, how many of them conduct verification of the reward models they use? Most of them do not even mention any details of the reward models they use. They just directly presume all these assumptions exist and give the scores as evidence of their alignment algorithms' effectiveness. This bad paradigm needs to be changed. We conduct experiments which may seem intuitive and do not lead to any surprising results, but we aim to provide a testbed and paradigm for verification of two-level human proxies used in further alignment research.
>
> **W2**. The authors have a wide range of empirical results - but it is not clear how the choice of models used for each experiment was made.
>
> **A2**: Thank you for pointing this out, we will provide more detailed model selection rationale in the revised version.
>
>
> [1] RRHF: Rank Responses to Align Language Models with Human Feedback without tears. NeurIPS 2023.
>
> [2] Scaling Laws for Reward Model Overoptimization. ICML 2023.

---

> ### Author Response · Authors · 2024-11-25
> **Response to Reviewer ZGTm (Part 2)**
>
> We would like to thank the reviewer for the insightful and constructive comments. The following are our detailed responses regarding all major concerns. We hope the following responses can clarify the missing points and address these concerns.
>
> **W3**. Starling-34B has the highest reward model accuracy (Table 4) and using Startling-34B as a reward model leads to a better model (Figure 6). Doesn’t this go entirely against the authors claim that current day reward models (level-2 proxy) are inadequate?
>
> **A3**: Thanks for asking, but we never made this wrong claim that "current day reward models are inadequate". In fact, people can find many reward models with good performance in RewardBench[1]. What we do claim is that current alignment works choose bad reward models as human proxies for alignment optimization and evaluation, and even fail to conduct verification before usage. For example, as shown in Figure 6, PRO is an alignment approach proposed by [2]. Pythia-1.4B is the bad reward model PRO used for alignment optimization. Its bad performance can be found in Table 3. Pythia-6.9B is used for alignment evaluation in PRO. Its bad performance can be found in Appendix, Table 8. Authors of PRO use these bad Level-2 proxies without any verification. In Figure 6, we instead turn to use Starling-34B, a verified good human proxy, and achieve much better alignment performance. Therefore, "Starling-34B has the highest reward model accuracy (Table 4) and using Starling-34B as a reward model leads to a better model (Figure 6)" actually strengthens our argument. We aim to raise researchers' attention to choosing trustworthy human proxies (line 496-501) and highlight the need for proper verification before usage.
>
> **W4**: Table 4: I don’t think assessing DPO for reward modeling is reasonable. The reason I mention this is that if you take out the DPO results in Table 4, the claims that the authors make (ie, reward models often do worse than 50/50 chance) is significantly weakened.
>
> **A4**: We are sorry for the misunderstandings caused here. We indeed compute the implicit reward that the DPO derivation uses (ratio of log probs) in Table 4 for DPO results. Even with DPO results taken out, results of GPT-J and Beaver-7B still support our claim that some reward models perform even worse than random. Moreover, there are other reward models that support our claim, such as DeBERTa-RM [3], another widely used reward model in alignment works. We don't put all the results here due to the consideration of space limitations and consistency with later experiments. We will add all the results in the Appendix in the revision. Thanks for your reminder.
>
> ## Addressing Questions:
>
> **Q1**. Some details on HH-RLHF would be helpful.
>
> **A1**: You can find detailed information in Appendix A.
>
> **Q2**. Figure 4 – why did you include GPT-J? Also, how big of a model is GPT-J?
>
> **A2**: Thanks for pointing this out, we will give a more clear explanation in the revised version. GPT-J (6B) is a reward model used as an automatic evaluator of alignment performance in RRHF[4]. We use it to show its bad correlation with human evaluation and demonstrate why it should not be used as an evaluator, which also casts doubt on the trustworthiness of RRHF's results and effectiveness.
>
> **Q3**. Figure 4 - why is there no correlation score?
>
> **A3**: Thank you for pointing this out. We will add the correlation score in the revised paper.
>
> **Q4**. Does the newly labeled data include annotations from all 4 annotators? If you have already gone through the effort to re-label the data, having this more granular information could allow for better alignment methods as opposed to binary (preferred vs. rejected) labels.
>
> **A4**: Yes. We agree with you that preference data with more granular information may facilitate more alignment algorithms. However, as we have repeatedly conveyed in our paper, we should not drain every effort in chasing alignment performance, especially in current research literature, where untrustworthy human proxies are widely adopted for alignment optimization and evaluation.
>
> **Q5**. What is the purpose of Figure 5?
>
> **A5**: This figure mainly illustrates that reward scores during training don't correlate with final alignment performance. Lines 450-463 have explained this in detail.
>
> [1] https://huggingface.co/spaces/allenai/reward-bench
>
> [2] PRO: Preference Ranking Optimization for Human Alignment. AAAI 2024.
>
> [3] https://huggingface.co/OpenAssistant/reward-model-deberta-v3-large-v2
>
> [4] RRHF: Rank Responses to Align Language Models with Human Feedback without tears. NeurIPS 2023.

---

### Official Review · Reviewer_c3YA · 2024-11-03

**Soundness:** 3
**Presentation:** 2
**Contribution:** 3
**Rating:** 6
**Confidence:** 4

**Summary:**

This work explores the various complexities of alignign to human preferences, considering both the more direct capture of human feedback data as a preference proxy, and the one step removed reward model representation of human preference. It provides various experimental insights into how faithfully these reflect actual human preference. The authors analyze and provide a cleaned version of HH-RLHF, and use this as a basis for further analysis.

**Strengths:**

The high opposite re-label rates for HH-RLHF are insightful and highlight various issues with working with depending heavily on such datasets. In this setting, three of four annotators choose the rejected response as better, highlighting critical issues with the dataset.

The experimental design, discussion and framing are insightful. Experiments such as the correlation between human and model evaluations are particularly valuable.

**Weaknesses:**

The DPO experimental settings are clear, in terms of training on both the original HH-RLHF and improved CHH-RLHF datasets. However, this is unclear for the PPO setting and even on re-reading, I am unsure which results (if they exist) show the comparison between the effects of RMs trained on HH-RLHF vs CHH-RLHF on downstream LLM performance evals. If these experiments have not been run, I would strongly recommend running them as they would speak to the impact of the massive effort to re-label/clean up this data. If this is not possible due to resource constraints, I'd recommend discussing this in detail.

Overall, this is thorough work which is will motivated and has involved important and substantial effort, both in terms of the CHH-RLHF annotation and experimental design, much of which is extremely insightful. I feel that currently, the poor framing of the work within the context of existing critical investigations of the space and lack of clarity around the overall narrative limit the work's potential impact. Both these points can be addressed to make this a potentially high impact contribution.

**Questions:**

- grammar L084: If not, what impacts they may have on alignment performance?

---

> ### Author Response · Authors · 2024-11-25
> **Response to Reviewer c3YA**
>
> We would like to thank the reviewer for the insightful and constructive comments. The following are our detailed responses regarding all major concerns. We hope the following responses can clarify the missing points and address these concerns.
>
> ## Addressing Weaknesses:
>
> **W1**: No experiments on PPO with reward models trained on HH-RLHF vs CHH-RLHF.
>
> **A1**: We have run experiments on PPO according to your advice. We trained LLaMA2-7B-chat on HH-RLHF and CHH-RLHF respectively as our reward models. The experimental results are shown below.
>
> | Model | ACC on ${CHarmless}_{base}$ | ACC on ${CHelpful}_{base}$ | ACC on ${CTest}_{mixed}$ |
> |-------|----------|------|------|
> | RM trained on CHH-RLHF  | 78.68 | 73.92 | 71.60 |
> | RM trained on HH-RLHF | 67.18 | 65.84 | 63.95 |
>
> | Model | RM Score on ${CHarmless}_{base}$ | RM Score on ${CHelpful}_{base}$ | RM Score on ${CTest}_{mixed}$ |
> |-------|----------|------|------|
> | PPO + RM trained on CHH-RLHF  | -7.9 | -7.8 | -7.8 |
> | PPO + RM trained on HH-RLHF | -8.1 | -8.3 | -8.0 |
>
> From the results, we can see that CHH-RLHF serves as a better Level-1 proxy, which leads to a better Level-2 proxy. Moreover, PPO using a better Level-2 proxy within the constraint of Goodhart's law achieves better alignment performance. This experiment further emphasizes the importance of the trustworthiness of human proxies. We will add a more thorough version of this experiment in the revision according to your advice.
>
> **W2**: Framing and narrative problems.
>
> **A2**: Thank you very much for your kind and insightful advice. We will fix these problems in the revised version.
>
> ## Addressing Questions:
>
> **Q1**: Grammar error in L084.
>
> **A1**: Thank you for pointing this out, we will fix this in the revised version.

---

> > ### Comment · Reviewer_c3YA · 2024-11-26
> >
> > Thank you for running the additional results. These further demonstrate the value of the improvements made to the dataset. I have increased my score to 6.

---

### Official Review · Reviewer_Deme · 2024-11-04

**Soundness:** 4
**Presentation:** 4
**Contribution:** 2
**Rating:** 3
**Confidence:** 4

**Summary:**

This paper explores the challenges of using human-made proxies, like labelled data and reward models, to align large language models (LLMs) with human preferences. The authors find that these proxies often don’t accurately reflect genuine human values, which can lead to unintended model behaviours. Therefore, they re-labelled a popular preference learning dataset (HH-RLHF) to create a cleaner version, called CHH-RLHF, which they hope will support more reliable alignment. Their experiments show that better proxies lead to improved alignment, but they caution against relying too much on any single measure, as it can be over-optimized and lose accuracy. The authors urge researchers to prioritise verifying proxy reliability to prevent these hidden risks from undermining model alignment.

**Strengths:**

- The paper's classification of human proxies into two levels (direct preference data and reward models) offers a clear framework. This seems simple but is actually very **novel**. This is the first time I have ever seen such a classification.
- By re-labelling the HH-RLHF dataset and providing a cleaner version (CHH-RLHF), the authors make a very practical contribution that could improve the reliability of the preference learning dataset.
- By discussing Goodhart's Law, the paper raises awareness of the risks of over-optimizing proxies. This is indeed important for us to keep in mind: alignment isn’t just about maximising scores but ensuring genuine alignment with human intent.

**Weaknesses:**

- “This work aims to alert researchers to the importance of verifying human proxies before using them, whether for alignment optimization or evaluation.” Several research papers discuss this issue, with reward over-optimization being one of the most well-known. This topic has been widely discussed.

- Relabeling HH-RLHF is valuable, but the quality of the HH-RLHF dataset is somewhat criticized in the community. The more important question is how we can collect user preferences in a scalable and reliable manner rather than simply re-labelling them. This paper is valuable for providing a re-labelled dataset, but in my opinion, it does not meet the standard of ICLR.

- “However, they only establish a leaderboard for reward models, failing to reveal the relationship between alignment performance and reward model quality.” I agree, and it’s already been discussed in the community that achieving a higher score in RewardBench does not necessarily translate into better downstream task performance. How do the authors of this paper address this issue?

- Table 3 is not convincing, as Figure 4 already shows that “Starling-34B” provides a better score. Why not use 34B? Instead, a 6.9B model was chosen. I wouldn’t consider this a pitfall; experiments clearly indicate that a larger base model performs better.

- In Figure 5, all models are using 6.9B or 7B reward models: why not use the 34B version?

- As the authors have already empirically shown that a better reward model leads to a better policy model, where is the pitfall?

**Questions:**

- Where is the re-labeled preference dataset?

---

> ### Author Response · Authors · 2024-11-25
> **Response to Reviewer Deme**
>
> We sincerely thank you for your valuable and constructive comments. The following is our detailed responses regarding all major concerns.
>
> ## Addressing Weaknesses:
>
> **W1**: Novelty of research topic.
>
> **A1**: Research on this topic is still limited. Although OpenAI has explored the reward model over-optimization problem [1], "under-optimization" of reward models is still in researchers' blind zone. Reward models with poor performance used in alignment optimization and evaluation are quite common [2,3], and their negative effects are still underestimated. Therefore, in this work, we conduct extensive experiments to unveil pitfalls of two-level human proxies, including both reward models and preference data.
>
> **W2**: The more important question is how we can collect user preferences in a scalable and reliable manner rather than simply re-labelling them.
>
> **A2**: Relabeling the HH-RLHF dataset is not a key contribution in our work. We just use this commonly used dataset as an example to show severe pitfalls of Level-1 proxies in existing alignment research. The detection of Level-1 proxy pitfalls aligns with core alignment principles (harmlessness, honesty, and helpfulness), making the pitfall inventory scalable. Researchers can further develop methods for getting better preference data based on the inventory of pitfalls summarized in our paper.
>
> **W3**: It’s already been discussed in the community that achieving a higher score in RewardBench does not necessarily translate into better downstream task performance. How do the authors of this paper address this issue?
>
> **A3**: According to our experience, there is no clear connection between good reward model performance on some benchmarks and good alignment performance. Firstly, there is an upper bound (Goodhart's law) for reward models and alignment optimization. Secondly, based on our empirical exploration, we have found that reward models lacking robustness can perform good on some benchmarks but lead to suboptimal alignment performance. This is a very thought-provoking and interesting question that requires systematic exploration in the future.
>
> **W4**: Table 3 is not convincing, as Figure 4 already shows that "Starling-34B" provides a better score. Why not use 34B? Instead, a 6.9B model was chosen.
>
> **A4**: Table 3 shows the results of Pythia-6.9B, which is used in PRO[2] for alignment evaluation. We aim to use this work as an example to reveal that the reward model is quite inaccurate and should not be used in alignment without verification, which has unfortunately happened. Using such a poor reward model also casts doubt on the effectiveness of PRO.
>
> **W5**: In Figure 5, all models are using 6.9B or 7B reward models: why not use the 34B version?
>
> **A5**: We conduct experiments on four models of the same size to get scale-independent results, facilitating analysis of fundamental issues. There are not many 34B reward models available for us to experiment with. Besides, since this experiment explores the effects of reward models on alignment performance during training, we found it unnecessary to use 34B reward models.
>
> **W6**: As the authors have already empirically shown that a better reward model leads to a better policy model, where is the pitfall?
>
> **A6**: Exactly, pitfalls can be avoided by choosing trustworthy human proxies. However, in current literature, researchers fail to use trustworthy human proxies, which may cause pitfalls such as misalignment or reintroducing toxicity. A vivid example is PRO, as shown in Table 3. Therefore, our work aims to draw attention to the trustworthiness of human proxies and advocate for conducting verification before usage.
>
> ## Addressing Questions:
>
> **Q1**: Where is the re-labeled preference dataset?
>
> **A1**: We will open-source the CHH-RLHF dataset upon acceptance.
>
>
> [1] Scaling Laws for Reward Model Overoptimization. ICML 2023.
>
> [2] PRO: Preference Ranking Optimization for Human Alignment. AAAI 2024.
>
> [3] RRHF: Rank Responses to Align Language Models with Human Feedback without tears. NeurIPS 2023.

---

### Official Review · Reviewer_MLpc · 2024-11-06

**Soundness:** 2
**Presentation:** 2
**Contribution:** 1
**Rating:** 3
**Confidence:** 4

**Summary:**

The paper claims to be a an analysis of the role of human preference proxies (direct preferences and reward models) in LLM alignment. The authors do qualitative analyses of the errors in a commonly used alignment dataset and re-label it. They then show that this leads to a significant gain in alignment scores for both DPO style algorithms and PPO-style reward model + policy learning.

**Strengths:**

It is true that practitioners systematically under-account for the effect of annotation noise in preference-based alignment / RLHF. annotated data is frequently taken as ground truth, and there is less data cleaning performed than there should be.

The revised dataset could be an additional resource for the community.

**Weaknesses:**

After having gone through the paper, it seems the main contribution really boils down to a re-labeling of the (widely used) HH-RLHF dataset. While this will make it a useful resource, it is unclear to me what the scientific value of the work is. It is entirely expected that re-labeling the dataset to reduce noise, should improve accuracy on the downstream tasks.

What further insights are gleaned by this exercise? There is an inventory of the data labeling losses which are mildly interesting. However, the authors give absolutely no information about the labeling process. So what can a reader takeaway that is useful? let's grant that the downstream results indicate that the labeling process of this paper is better than the labeling for the original dataset, how do we know why?
Was the entire dataset re-labeled by the authors? this is not a process that can scale to other problems, were lower quality raters still need to be used. The inventory of "pitfalls" may be helpful but it is unclear whether the categories and proportions will generalize to other datasets/ domains.

The discussion section 5 does not seem to add any new insights not familar in the literature. 5.1 is more of a related work section (very minimal) and 5.2 goodharts law is well known and not quite relevant to the work done here.

This paper feel like a better fit for a conference like Findings'EMNLP.

**Questions:**

line 239: DPO is not really an SFT method. it is a simplfied form of RLHF.

line 365: should it be such as 1-0 metric, or something that takes into account the magnitude of the difference between the rewards.

line 377: for DPO, are you computing the impliicit reward that the DPO derivation uses (Ratio of log probs) ?

line 455 - 463: This finding if it can be replicated in other settings would be quite interesting (that reward accuracy during training is not reliable).

---

> ### Author Response · Authors · 2024-11-25
> **Response to Reviewer MLpc (Part 1)**
>
> We sincerely thank you for your valuable and constructive comments. The following is our detailed responses regarding all major concerns.
>
> ## Addressing Weaknesses:
>
> **W1**: Scientific values and further insights of this work.
>
> **A1**: The key of our work, as indicated by the title, is to expose the pitfalls of two-level human proxies in the alignment process, rather than simply relabeling a dataset. This is more of a criticism paper for existing alignment research, not one that proposes a dataset or an alignment method. As we have repeated in our paper, this work aims to serve as an alarm bell and arouse researchers' attention to the trustworthiness of two levels of human proxies. Our contribution goes beyond just dataset re-labeling. We highlight fundamental issues in alignment research that have been overlooked. Specific scientific values and insights of this work include:
>
> (1) First systematic categorization of human proxies into two levels;
>
> (2) Presentation of empirical evidence showing how proxy quality affects alignment performance and demonstrates how inferior proxies can lead to misalignment;
>
> (3) Critical analysis of reward model reliability in evaluation;
>
> (4) Provides empirical evidence of disconnect between reward model scores and actual alignment quality during training;
>
> (5) Offers a testbed and practical guidelines for further alignment research to verify proxy quality.
>
> **W2**: Authors give absolutely no information about the labeling process.
>
> **A2**: We give labeling information in Section 3. Specifically, we detected and concluded pitfalls in HH-RLHF dataset and fix them. For example, we have found toxicity in some chosen responses in the dataset. Lines 159-160 and 183 illustrate how the "Toxic" data is labeled. Labeling information can be found in corresponding parts for other data with pitfalls. But we feel that this type of illustration may be scattered for readers who hope to quickly find all the labeling information. Thank you for pointing out this problem. We will add a section for the complete labeling process in the Appendix in the revised version.
>
> **W3**: Was the entire dataset re-labeled by the authors?
>
> **A3**: No, we hired annotators to perform the re-labeling. Detailed annotator information, including their qualifications and the annotation process, can be found in Appendix B.
>
> **W4**: Scalability and generalization of labeling and pitfall categories and proportions to other datasets or domains.
>
> **A4**: The detection of Level-1 Proxy pitfalls aligns with core alignment principles (harmlessness, honesty, and helpfulness), making the labeling and pitfall inventory scalable.
> Previous datasets had such issues, partly due to being unaware of these problems. For example, during the annotation process of the HH-RLHF dataset, annotators weren't specifically reminded to pay attention to the presence of toxic content.
> While proportions might vary, the identified pitfall categories are fundamental issues that would apply to other preference datasets and domains. Therefore, the inventory of pitfalls for Level-1 Proxy in our paper could help improve existing labeling processes and inform better dataset creation practices.
>
> **W5**: Section 5.1 is a very minimal related work section.
>
> **A5**: The minimality of this part is due to limited research about the trustworthiness of human proxies in current literature.
>
> **W6**: Goodhart's law is well known and not quite relevant to the work done here.
>
> **A6**: Another reviewer has acknowledged our discussion on Goodhart's law as a strength: "By discussing Goodhart's Law, the paper raises awareness of the risks of over-optimizing proxies. This is indeed important for us to keep in mind: alignment isn't just about maximizing scores but ensuring genuine alignment with human intent." (Reviewer Deme).
> In fact, this work serves as a complementary study to OpenAI's work [1]. Since OpenAI's work primarily explores the over-optimization problem of level-2 proxies by introducing Goodhart's law into alignment, our work correspondingly discusses the under-optimization problem of level-2 proxies. Together, these works establish the upper and lower bounds of alignment optimization.
> We are grateful for your question that makes us realize that our writing may not be clear. We will fix this problem in the revised version.
>
> [1] Scaling Laws for Reward Model Overoptimization. ICML 2023.

---

> > ### Comment · Reviewer_MLpc · 2024-11-26
> > **a few points in response.**
> >
> > A2:
> >
> > Thank you , sorry i missed that in the first read.
> >
> > A5: The minimality of this part is due to limited research about the trustworthiness of human proxies in current literature.
> >
> > Some references:
> >
> > https://arxiv.org/html/2405.18952v1
> > https://arxiv.org/pdf/2311.04919

---

> ### Author Response · Authors · 2024-11-25
> **Response to Reviewer MLpc (Part 2)**
>
> We sincerely thank you for your valuable and constructive comments. The following is our detailed responses regarding all major concerns.
>
> ## Addressing Questions:
>
> **Q1**: Inaccurate writing about the DPO method.
>
> **A1**: Thank you very much for pointing this out. We will clarify in the revised version that DPO is a simplified form of RLHF that doesn't require explicit reward modeling, rather than strictly being an SFT method.
>
> **Q2**: Should it be such as 1-0 metric, or something that takes into account the magnitude of the difference between the rewards.
>
> **A2**: Yes, improving the metric would provide more nuanced understanding of reward model performance. But we just use this simple metric here to illustrate the severe inaccuracy of existing reward models used for alignment optimization and evaluation. Therefore, we only adopt this commonly used metric in our work to get general knowledge of reward model quality.
>
> **Q3**: Are you computing the implicit reward that the DPO derivation uses (Ratio of log probs)?
>
> **A3**: Yes.
>
> **Q4**: Further exploration on reward unreliability during training.
>
> **A4**: Thank you for your insightful advice. We will add more analysis here, which may highlight another potential pitfall in current alignment practices.

---

### Comment · Reviewer_ZGTm · 2024-11-26

Dear authors,

Thank you for your responses.
It seems as though my review + questions + conclusions are similar to that of MLpc and Deme, and thus I figured I'd write an overall comment.

From your response, you emphasize that the main contribution you wish to make is in raising awareness regarding the pitfalls in contemporary RLHF developments. However, the takeaway that all 3 of us had was that the main contribution of the work is in re-labeling HH-RLHF. I think this speaks to how the paper currently reads, and would suggest thinking about how to better emphasis the "alarm bells" as the takeaway instead.

I think the authors can take the questions from the reviewers as suggestions on how to do so. For instance, perhaps by including a discussion on the data collection process -- how to collect user preferences in a scalable and reliable manner moving forward -- which is mentioned by both MLpc and Deme.

Another common takeaway from reviewers is that the authors empirically demonstrate that larger reward models lead to better aligned models -- to which most reviewers responded that there is no surprise here, and so what exactly is new here and what is the pitfall?  I think most researchers/developers likely know this and want to use larger reward models if their budget allows.
It seems that the authors were trying to use this empirical finding to point out that the community should be more conscious when choosing their reward models, which is a valid point -- however, the fact that most reviewers shared the same response again speaks to how the paper currently reads.

Most importantly, all 3 of us have asked something along the lines of "What is the scientific value / further insights of this work" - to which the authors enumerate their contributions (response to MLpc), but personally I don't think the list of contributions answers this question. Perhaps because again, we each mention that we all know that cleaner data leads to better models, and better reward models lead to better aligned models -- and this work's enumerated list of contributions empirically demonstrate this. So then what further scientific insight should we take away from this?

Though I did not mention this, to be honest I had the same reaction to MLpc upon originally reading this work, which is that the paper seems to be a better fit for Findings of EMNLP/ACL.

---

> ### Comment · Reviewer_MLpc · 2024-11-26
> **concurrence**
>
> I would like to concur with everything this reviewer has said. I do want to encourage practitioners to focus on role of data quality and how it relates to downstream model performance. However there is little scientific insight in this paper, that would be useful for an ICLR audience. I encourage authors to think about these questions for their next iteration.

---

### Meta-Review · Area_Chair_rLvV · 2024-12-24

**Metareview:**

The paper investigates the reliability of human preference proxies, categorized as Level-1 (preference data) and Level-2 (reward models), in aligning large language models (LLMs). The authors re-label a widely-used dataset (HH-RLHF) to create a cleaner version (CHH-RLHF), leading to significant alignment improvements.

Strength:
This paper identifies the quality issue in HH-RLHF dataset and relabel them to improve the quality.

Weakness:
The major weakness pointed out by several reviewers is that the paper does not deliver additional scientific insights. It is well known that better data quality will lead to better reward model, which further leads to better aligned model.

Overall, the paper has limited contributions and is an over claim of the contribution. The issues or pitfalls raised by the paper is already recognized by the community. However, the paper does not provide insights or solutions that are scalable and generalizable. Therefore, the paper cannot be accepted as is.

**Additional Comments On Reviewer Discussion:**

During the rebuttal phase, the authors provide responses to the review feedbacks, including the major concern on the key scientific value and contribution of the paper. However, the response still does not address the concerns here. In particular, reviewers still thinks that the paper does not provide additional insights or scientific value other than relabeling the existing HH-RLHF dataset, which people in the community know that it has some quality issue. And also the claims and pitfalls mentioned in the paper is also well recognized as well and the rebuttal does not change it.

---

### Decision · Program_Chairs · 2025-01-22

Reject